# ON THE UNIVERSALITY OF NEURAL ENCODINGS IN CNNS

**Florentin Guth**
New York University & Flatiron Institute
florentin.guth@nyu.edu

**Brice Ménard**
Johns Hopkins University
menard@jhu.edu

## ABSTRACT

We explore the universality of neural encodings in convolutional neural networks trained on image classification tasks. We develop a procedure to directly compare the learned weights rather than their representations. It is based on a factorization of spatial and channel dimensions and measures the similarity of aligned weight covariances. We show that, for a range of layers of VGG-type networks, the learned eigenvectors appear to be universal across different natural image datasets. Our results suggest the existence of a universal neural encoding for natural images. They explain, at a more fundamental level, the success of transfer learning. Our work shows that, instead of aiming at maximizing the performance of neural networks, one can alternatively attempt to maximize the universality of the learned encoding, in order to build a principled foundation model.

## 1 INTRODUCTION

Deep neural networks reliably achieve high performance on visual tasks such as image classification, with remarkable robustness to the exact details of the architecture, initialization, and training procedure. Furthermore, transfer learning results show that, in some cases, networks trained on one task can perform well on other tasks by simply retraining the last few layers. In the context of computer vision tasks, this raises the question: to what extent do networks share a universal encoding of images, irrespective of their architecture and training dataset? What does this encoding look like? Can one identify generic learned features across datasets and tasks?

Deep networks can be studied through two main points of view: one can compare their representations (how an input image is expressed through neural activations) or their *encodings* (how a training dataset is encoded in network weights). The dominant approach to studying the properties of neural networks has been through representations. It has been shown that representations learned by networks trained from different initializations appear to be similar over many layers (Raghu et al., 2017; Kornblith et al., 2019). Similar observations have been made in the context of human neural representations by studying fMRI response patterns in visual cortex (Haxby et al., 2011), as well as between neural network representations and IT spiking responses (Yamins et al., 2014). Here, we ask whether this similarity at the level of network representations arises from a more fundamental similarity between their learned weights, i.e., in the functions implemented by different networks rather than their outputs. In addition, we explore which aspects of the encoding are similar.

Neural encodings (or network weights) are less easily prone to analysis than hidden representations and have thus been less studied. The weights of convolutional neural networks (CNNs) learn to exploit simultaneous correlations across channels and between neighboring pixels to solve numerous tasks. The first layer of CNNs can be directly visualized and the learned spatial filters are known to be Gabor-like filters (Krizhevsky et al., 2012) for a wide range of settings. Recent work by Cazenavette et al. (2022) pointed out that, in CNNs, most of the learning deals with channel mixing: the spatial filters can be kept frozen from their random initializations, and the remaining learning of channel mixing leads to similar performance. Exploring the structure of all weights, including channel-mixing dimensions, is more challenging. While the input basis of the first layer is fixed (for images, it is aligned with the pixel coordinate system), the other layers are expressed in random bases set by the neurons of the previous layers. This random shuffling has been a challenge to explore the properties of encodings and, to a great extent, is at the origin of the black box qualification of neural networks.

Recently, Guth et al. (2023) have shown that this random shuffling can be canceled by aligning the hidden representations to a common reference.

In this paper, we explore the statistical properties of CNN weight tensors following the approach of Guth et al. (2023). We first find that the learned eigenvectors of spatial filters are surprisingly simple. They are low-dimensional and, to first order, they do not depend on the filter size, dataset or task over a wide range of settings. In other words, a universal set of spatial filter eigenvectors emerges. It is therefore possible to factorize CNN weight tensors to separate the information processing between space and channels. We do so using a frozen set of such spatial filters and only learn weights along channels. It then becomes possible to compare learned channel eigenvectors between networks and we show that, in the context of natural images, the learned channel-mixing features also show signatures of universality: they are similar across datasets well beyond the first few layers.

Our main contribution is to characterize the joint processing of CNNs along spatial and channel dimensions and show that they typically use a preferred encoding strategy:

- We develop a procedure to compare the weights of networks rather than their representations. This can be applied to measure similarities between neural encodings of different datasets and thus, indirectly, measure similarities between datasets.
- We show the emergence of a seemingly universal encoding of natural images, through both spatial filters and to a lesser extent channel weights. We find encoding similarity across datasets and tasks over an appreciable range of layers. It explains at a more fundamental level the success of transfer learning, self-supervised learning, and foundation models.

## 2 EXPLORING LEARNED SPATIAL FILTERS

### 2.1 SPATIAL FILTERS IN CNNS

At each layer, a convolutional neural network simultaneously mixes information across the spatial and channel dimensions of a representation by learning a weight tensor $W$ containing $C_{\text{out}} \times C_{\text{in}} \times k \times k$ trainable parameters. In general, the respective importance of spatial and channel mixing is unclear as the two are entangled in a single linear operation. How does the training process use this multi-dimensional capacity? How much and what kind of information emerges in the spatial and channel domains?

The first layer of CNNs, which can be directly visualized, is known to learn Gabor-like filters (Krizhevsky et al., 2012) in a wide range of settings. However, the properties of learned spatial filters at deeper layers have not yet been exposed clearly. To gain some insight, we first train a VGG-11 (Simonyan & Zisserman, 2015) (which has $L = 8$ convolutional layers) on the ImageNet dataset (Russakovsky et al., 2015). We first set the filter size $k$ to 7 to ease interpretation. We slightly simplify the architecture by removing biases and using a linear classifier, see Appendix B for details. A visual inspection of the spatial filters of each layer reveals a set of rather smooth and redundant spatial filters, suggesting a low-dimensional statistical description. A subset is shown in Appendix A.

### 2.2 SPATIAL EIGENVECTORS

To perform a statistical comparison, we compute eigenvectors of spatial filters at all layers for different filter sizes. More precisely, we diagonalize the $k^2 \times k^2$ "spatial" covariance obtained by averaging over the $C_{\text{out}} \times C_{\text{in}}$ channel dimensions as in Trockman et al. (2023). The resulting leading eigenvectors are shown in panel (a) of Figure 1. Interestingly, for layers $l \geq 2$, we observe very consistent sets of eigenvectors, dominated by a low-dimensional subset, which was not apparent on visual inspection of the individual filters. This means that the main difference between the covariances of the layers (as seen in Appendix A) originates from the eigenvalues rather than the eigenvectors. We also note a slight dilation of the filters affecting all eigenvectors with depth. On occasions, subsequent ranks appear in flipped order. This happens when consecutive eigenvalues are similar. Panel (a) also shows the eigenvectors of filters learned with smaller filter sizes $k = 3$ and 5. Although the resulting filters are in different spaces $\mathbb{R}^{3 \times 3}, \mathbb{R}^{5 \times 5}, \mathbb{R}^{7 \times 7}$, their visual correspondence is clear. It suggests that these eigenvectors are determined by a mechanism that does not depend on layer depth nor filter size. Pushing the exploration further, we show in panel (c) similar measurements for different

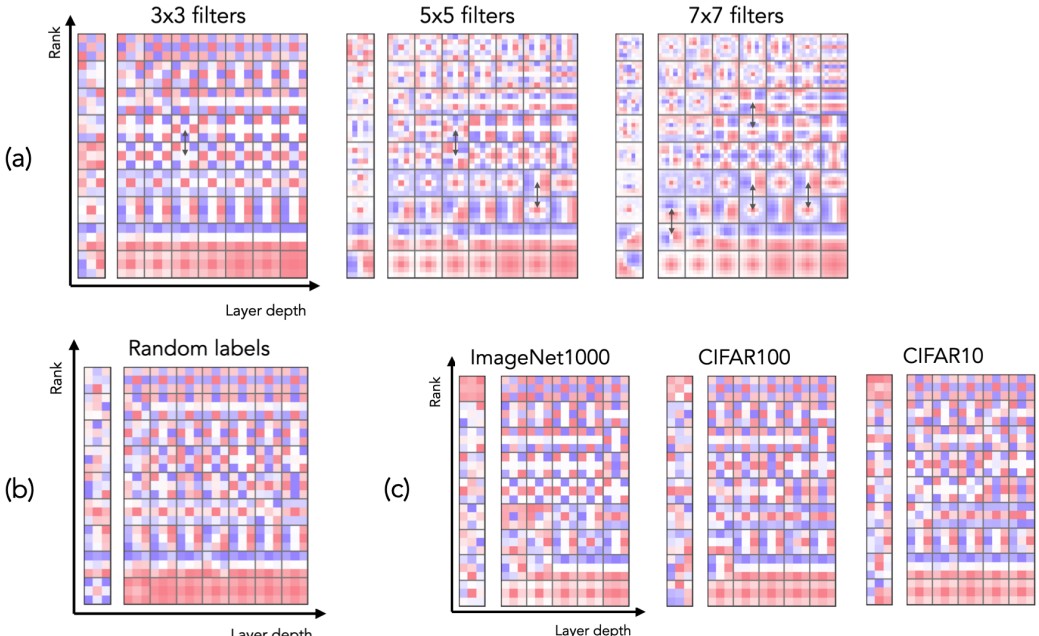

Figure 1: Visualization of spatial filter eigenvectors learned by VGG networks in various settings. *(a)* Learned spatial eigenvectors on ImageNet for different filter sizes. For larger filter sizes, only the first 9 eigenvectors are shown. The same set of eigenvector patterns can be seen for all layers $l \geq 2$, while the first layer (shown separately) displays a different behavior. On occasions, subsequent ranks appear in flipped order due to similar eigenvalues, as indicated with arrows. *(b)* When training with random labels on ImageNet, we recover the same set of filters, slightly distorted. The spatial eigenvectors can thus be mostly learned without labels. *(c)* Spatial eigenvectors for VGG networks trained on different datasets. The ImageNet dataset was resized to the $32 \times 32$ resolution for direct comparison with the other datasets. The learned spatial eigenvectors are similar across datasets (and depth), and are similar to the filters learned on higher-resolution images. Note that only the first 6 convolutional layers are relevant given the smaller size of the images.

datasets (CIFAR10, CIFAR100, and ImageNet) on standard classification tasks and resized to the same $32 \times 32$ resolution. In all cases, the eigenvectors show again the same canonical set. It is slightly noisier than the version presented above due to the lower resolution of the images. We also note that the effective spatial support of the filters is slightly smaller for lower-resolution images, as can be expected. Finally, we explore the effect of the training task: we train our network on ImageNet (at $224 \times 224$ resolution) on random labels (Zhang et al., 2021), which can be considered as a self-supervised training task leading to a compressed representation of the data. The corresponding eigenvectors, shown in panel (b), reveal that even with random labels, the same set of eigenvectors emerges again (Maennel et al., 2020). In summary, this canonical set is observed for different filter sizes, datasets and tasks: it shows signatures of universality.

Several researchers have proposed that certain aspects of the weights of CNNs may be determined by symmetry groups of the data (Cohen & Welling, 2016; Kondor & Trivedi, 2018; Marchetti et al., 2023). Both Trockman et al. (2023) with artificial networks and Pandey et al. (2022) in the context of receptive fields of biological sensory neurons have shown that the covariance of learned spatial filters can be mathematically modeled. In summary, the existence of a canonical set of spatial filters which can be defined mathematically implies that most of the learning ends up taking place along channel dimensions.

## 2.3 FACTORIZING SPACE AND CHANNELS

Motivated by the existence of a canonical set of spatial filters, we introduce a simplified architecture: we factorize the standard 2D convolution operation into a depthwise and pointwise ($1 \times 1$) convolution

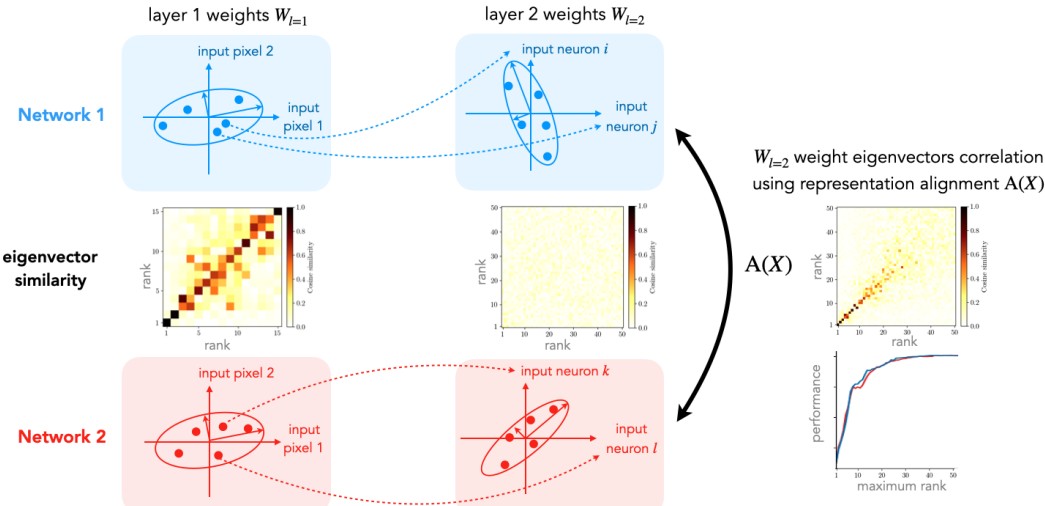

Figure 2: A schematic view of the first two layers of two networks trained from different random initializations. At layer one, the weight eigenvectors are aligned by default as the input originates from the aligned image pixels. *Collectively*, the neurons act as an operator filtering certain directions of variation. *Individually*, each neuron defines an axis for the next layer. Expressed in this random basis, layer-two weight eigenvectors are no longer aligned between the two networks. An activation-based representation alignment can be used to meaningfully rotate one basis onto the other. The middle panels show the cosine similarities between weight covariance eigenvectors of the two networks trained on CIFAR10, with and without activation-based alignment. The bottom-right panel shows that almost all the performance originates from the range of ranks for which the correlation is detected.

so that each layer operates only across spatial or channel dimensions, similar to separable architectures (Sifre & Mallat, 2013; Chollet, 2017). The depthwise convolution convolves each channel with a fixed set of frozen (non-learned) filters, which we set to be the universal spatial eigenvectors observed in the previous section. The non-linearity is applied after the depthwise convolution so that there is no linear combination of these spatial filters across channels. It is followed by the pointwise convolution, which is simply a $1 \times 1$ convolution, i.e., a linear operation along channels applied at each spatial location. Interestingly, this simplified architecture can recover most of the network performance, similarly to Cazenavette et al. (2022); Trockman et al. (2023). See Appendix B for more details on this architecture.

## 3 EXPLORING LEARNED CHANNEL WEIGHTS

Our goal is now to investigate what is being learned in the channel domain, using the factorized architecture introduced in the previous section. We explain in Section 3.1 how this can be reduced to comparing covariances of channel weights through an alignment procedure. We then define in Section 3.2 measures of dimensionality and similarity of these learned covariances. Finally, we use this approach to characterize neural encodings of different datasets and evaluate their universality in Section 3.3.

### 3.1 COMPARING CHANNEL WEIGHTS OF DEEP NETWORKS

How does one meaningfully compare (channel) weights of two trained deep networks? Here, we briefly review the approach introduced by Guth et al. (2023) and present the main ideas. The analysis is more challenging than for the spatial filters: the latter are expressed in fixed spatially-*aligned* axes, while channel weights are expressed in random bases set by the (randomly initialized) neurons of the previous layer. This requires an alignment procedure between hidden *representations* before comparing the weights. These concepts are illustrated in Figure 2.

**Aligning hidden layers and weights.** We begin by comparing two hidden representations $\phi(x)$ and $\phi'(x)$ learned by two different networks. In general, $\phi(x)$ and $\phi'(x)$ are not directly comparable (they might even have different dimensionalities). Representational similarity analysis (Kriegeskorte et al., 2008) instead compares their similarity structures (or kernels) $\langle \phi(x), \phi(y) \rangle$ and $\langle \phi'(x), \phi'(y) \rangle$, which have empirically been found to be close in various settings (Raghu et al., 2017; Kornblith et al., 2019). This implies that the variability in the representation between $\phi$ and $\phi'$ must preserve this similarity structure, and is thus limited to an orthogonal transform. In other words, when $\langle \phi(x), \phi(y) \rangle \approx \langle \phi'(x), \phi'(y) \rangle$, there exists an orthogonal alignment matrix $A$ such that $\phi'(x) \approx A\phi(x)$ (Guth et al., 2023). It is defined by minimizing the mean-squared error

$$\min_{A^{\mathrm{T}} A = \mathrm{Id}} \mathbb{E}\Big[\big\| A\,\phi(x) - \phi'(x) \big\|^2\Big], \tag{1}$$

over orthogonal matrices $A$. This is also known as the Procrustes problem (Hurley & Cattell, 1962) and can be solved in closed form (Schönemann, 1966). It corresponds to a so-called "shape metric" (Williams et al., 2021) on the kernels defined by the representations (Harvey et al., 2023).

Now consider two neurons $w$ and $w'$ in the next layer of the two different networks. What does it mean for $w$ and $w'$ to be equivalent? It seems natural to ask that the two neurons compute similar outputs: $\langle w, \phi(x) \rangle \approx \langle w', \phi'(x) \rangle$. Because $\phi(x) \neq \phi'(x)$, this condition is not equivalent to $w \approx w'$. Rather, using the fact that $\phi'(x) \approx A\phi(x)$, we have

$$\langle w', \phi'(x) \rangle \approx \langle w', A\phi(x) \rangle = \langle A^{\mathrm{T}} w', \phi(x) \rangle, \tag{2}$$

so that the two neurons compute similar outputs when $w \approx A^{\mathrm{T}} w'$, or equivalently, when $w' \approx Aw$. Just like the alignment $A$ maps representations in the first network to representations in the second network, it maps next-layer neurons in the first network to equivalent neurons in the second network. Comparing hidden neurons from different networks thus requires aligning their hidden representations and taking this alignment into account in the comparison.

**Comparing weight distributions through covariances.** Comparing individual neurons in two different networks amounts to searching for a one-to-one mapping between them. If the two networks had the same neurons, but possibly in a different order, then their representations would differ by a permutation (Entezari et al., 2022; Benzing et al., 2022; Ainsworth et al., 2022). The use of rotations when aligning representations suggests that more variability might be present. Rather than comparing individual neurons from two different networks, we search for similarities between the neural populations at a global level, i.e., whether they have the same statistics. This corresponds to testing whether the neurons in both networks can be modeled as samples from the same distribution, as done in so-called "mean-field" analyses of neural networks.

Which statistics of the neural populations should we then measure and compare? Guth et al. (2023) have shown that the covariance of neuron weights captures most of the encoding properties, as knowledge of the weight covariances can be sufficient to generate new networks with similar performance. In particular, the covariances at the end of training are the sum of a "learned" component and a "random" component, the latter mostly resulting from the initialization. The learned component is spanned by the leading eigenvectors and is rather low-dimensional, as low-rank approximations of the learned weights result in negligible performance loss (see Figure 2).

**Summary.** Though the overall behavior of the network only depends on the collective properties of the neurons, their weights are expressed in a basis defined by the individual neurons of the previous layer. To compare weights between two networks, for each layer, we thus compute the alignment matrix $A$ between the input representations and use it to align the neuron weights of both networks. The learned encodings can then be characterized by the leading eigenvectors (and eigenvalues) of the weight covariances.

## 3.2 MEASURING AND COMPARING WEIGHT COVARIANCES

Similarly to our approach for spatial covariances in Section 2.2, we can compare the eigenvectors of two channel covariance matrices after alignment. These eigenvectors however cannot be easily visualized and interpreted, though can be compared by measuring cosine similarities between them.

Another challenge is that channel covariance matrices are significantly higher-dimensional than spatial covariance matrices, and the number of neurons (which here correspond to samples) is comparable to or even smaller than the dimension. This results in inconsistent estimations of the covariance matrices: in particular, their eigenvalues and eigenvectors deviate significantly from their "ground-truth" values (if the number of neurons were infinite). Assessing whether two networks have the same channel covariances thus requires some care. The dimensionality and similarity measures of weight covariances we now introduce thus rely on a shrinkage of the covariance eigenvalues.

**Eigenvalue shrinkage.** The decomposition of the weight covariance in "learned" and "random" components corresponds to the so-called "spiked" covariance model (Johnstone, 2001), under which only eigenvectors with sufficiently large eigenvalues can be estimated (Baik et al., 2005). The learned component of the covariance is thus optimally estimated by shrinking the eigenvalues of the empirical covariance matrix (Donoho et al., 2018). More precisely, for a weight matrix of size $C_{\text{out}} \times C_{\text{in}}$, let $\gamma = C_{\text{in}}/C_{\text{out}}$, which measures the "sampling noise". The eigenvalues of the $C_{\text{in}} \times C_{\text{in}}$ weight covariance matrix are shrunk according to the formula

$$\lambda \longmapsto \begin{cases} \frac{\lambda-1-\gamma}{2} + \sqrt{(\frac{\lambda+1-\gamma}{2})^2 - \lambda} & \text{if } \lambda > (1 + \sqrt{\gamma})^2, \\ 0 & \text{otherwise,} \end{cases} \tag{3}$$

with a normalization so that the initialization variance is equal to 1. This removes most of the noise resulting from the limited number of neurons. We can thus meaningfully compare aligned weight covariances through their estimated "learned" components.

**Dimensionality of covariances.** A first, indirect way to compare the resulting learned covariances is by comparing their dimensionality. We define the effective rank $r_{\text{eff}}$ of a matrix as its eigenvalue-weighted mean rank. More precisely, if the covariance eigenvalues are $\lambda_1 \geq \cdots \geq \lambda_d$ after shrinking, we have

$$r_{\text{eff}} = \frac{\sum_{k=1}^{d} \lambda_k \, k}{\sum_{k=1}^{d} \lambda_k}. \tag{4}$$

We will see that this dimensionality measure is correlated with the difficulty of task (e.g., the number of classes).

**Similarity of covariances.** A more direct and fine-grained comparison can be done by comparing individual eigenvectors, as in Section 2.2. However, this suffers from instabilities, e.g. when individual eigenvalues are too close. A more quantitative, summarized measure of similarity may be desired, which can also more easily reveal trends as a function of hyper-parameters.

First, we wish to quantify the level of similarity between two covariance matrices $C_1$ and $C_2$ after eigenvalue shrinkage. A relevant metric is the Bures-Wasserstein distance (Bhatia et al., 2019), which corresponds to the optimal transport distance between two centered Gaussian distributions of respective covariances $C_1$ and $C_2$ (Peyré & Cuturi, 2019, Remark 2.31). It thus measures the displacement of individual neuron weights that is needed to change their global covariance from $C_1$ to $C_2$. It allows us to compare neural encodings by accounting for their statistical shapes and overlap. This distance is defined as $d^2(C_1, C_2) = \operatorname{tr} C_1 + \operatorname{tr} C_2 - 2\|C_1^{1/2}C_2^{1/2}\|_1$, where $\|\cdot\|_1$ is the nuclear norm (the sum of the singular values). It can be turned into a cosine similarity with

$$\cos \theta(C_1, C_2) = \frac{\|C_1^{1/2}C_2^{1/2}\|_1}{\sqrt{\operatorname{tr} C_1 \, \operatorname{tr} C_2}}. \tag{5}$$

This metric allows quantifying similarities at the level of covariances rather than individual eigenvectors and is therefore better suited at revealing correlations potentially diluted over ranks. For meaningful comparisons across varying layer dimensionalities, we normalize the resulting value. Its zero point is defined as the cosine similarity $\cos \theta(C_1, O\,C_1\,O^{\text{T}})$ between $C_1$ and a random rotation $O \in O(\mathbb{R}^d)$ of it, leading to non-zero but minimal eigenvector correlations. The upper bound is defined as the cosine similarity $\cos \theta(C_1, \widehat{C}_1)$ after a "resampling" of the covariance $C_1$. We define $\widehat{C}_1$ as the estimated (shrunk) covariance obtained from $C_{\text{out}}$ random Gaussian vectors of covariance $C_1 + \sigma^2\text{Id}$, where $\sigma^2\text{Id}$ is the initialization variance. This leads to the normalized cosine similarity

$$\text{S}(C_1, C_2) = \frac{\cos \theta(C_1, C_2) - \cos \theta(C_1, O\,C_1\,O^{\text{T}})}{\cos \theta(C_1, \widehat{C}_1) - \cos \theta(C_1, O\,C_1\,O^{\text{T}})}. \tag{6}$$

This normalization enables comparisons of similarity levels at different layers, and can thus be used to reveal trends across depth.

### 3.3 Universality of weight eigenvectors learned from natural images

We now compare the properties of neural encodings for networks trained on different datasets and tasks. To do so, we measure the absolute value of the cosine similarity between eigenvectors of the (aligned) weight covariances for the same factorized VGG architecture with frozen spatial filters but different training contexts. We present our results in Figure 3. Each panel shows encoding similarities of a given convolutional layer as a function of rank. We consider different training datasets and use subsets of CIFAR10, CIFAR100, and ImageNet all downsampled to $32 \times 32$ resolution. We use different tasks: true labels or random labels. The suffix (bis) refers to a different random initialization. Using the same image resolution and architecture for all datasets ensures that all networks have the same number of layers and receptive field sizes, allowing meaningful comparisons. For each trained network, the effective rank $r_{\text{eff}}$ is indicated by an arrow overlaid on the rank axis.

We begin our exploration by considering two networks trained on ImageNet but with different random initializations. The encoding similarities are shown in the top row of the figure. First, we can observe that, as depth increases, layer-based encodings become higher-dimensional. In addition, the eigenvector similarities shown as a function of ranks show that the learned channel eigenvectors of the two networks are very similar. This similarity will be quantified below. This comparison, for which only the random initialization changed, provides a reference for our next experiments. This is the maximum level of similarity one can expect.

We now consider networks trained on different datasets: CIFAR5 is the subset of CIFAR10 composed of the first 5 classes and ImageNet100a/100b are two randomly chosen but disjoint subsets of ImageNet classes. We present the encoding similarities in the bottom block of the figure. Interestingly, within the relevant ranks, we find a high level of weight eigenvector similarity for all pairs of training sets and over an appreciable range of layers. We can observe that datasets with more classes or more diversity (such as ImageNet100 compared to CIFAR100) lead to encodings with more relevant eigenvectors. Deeper into the networks, these similarities gradually vanish. This is expected as the encodings have to become task-specific toward the final classifier. This overview shows that, over a range of layers, networks learn a universal and relatively low-dimensional encoding of natural image datasets. This implies that, in principle, such encodings do not need to be learned and could be originally preset in a given architecture. Only the deeper task-dependent layers encodings would have to be learned. Figure 3 suggests that, for a range of layers, different trained networks provide us with different samplings of what could be a universal encoding of natural image datasets. These results reveal the origin of transfer learning capabilities.

We then consider networks trained on different classification tasks: using true or random labels with ImageNet downsampled to $32 \times 32$ resolution. For these experiments, we use data augmentation with random labels: it thus encourages the network to be invariant to horizontal flips and small spatial shifts while compressing the training dataset. The results are presented in the top block of the figure. Interestingly, for our VGG-type architecture, the weak correlation between the encodings of true and random label classification tasks reveals that the corresponding encoding strategies are significantly different. However, when comparing different realizations of random labels ("a" and "b"), one finds a common encoding scheme over a range of layers.

This overview of encoding similarities is summarized in Figure 4. It shows the normalized cosine similarity between the covariances of two trained networks (eq. 6), allowing a more quantitative comparison. In the left panel, we start by showing the encoding similarity for two networks trained on ImageNet but with different random initializations (blue curve). We then show the encoding similarities in the case of random labels (red and green curves). Similarly, we observe a high level of similarity up to the layer $\ell = 5$, showing that learning on different sets of random labels leads to similar encodings. In contrast, as mentioned above, when we compare what is learned with true and random labels, we find a much lower correlation: the two encodings are mainly different. We detect a subset of shared eigenvectors leading to a correlation amplitude of about $0.2$, showing the existence of some channel eigenvectors systematically learned in these two tasks. In the right panel, we show the encoding similarity for all of the pairs of datasets considered above. It reveals a continuous trend with depth and shows the existence of a shared encoding at most layers. In summary, our exploration

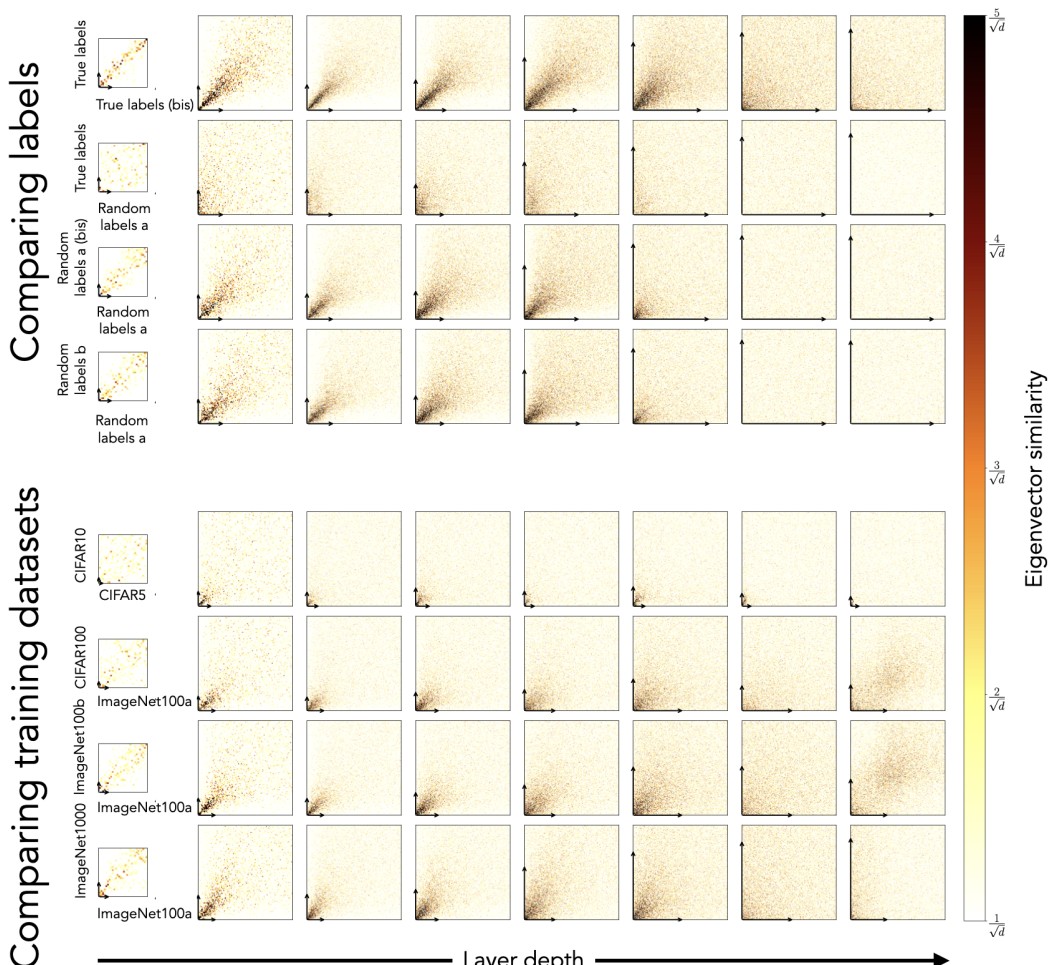

Figure 3: Universality of the leading covariance eigenvectors. Each panel shows pairwise cosine similarities between the first weight covariance eigenvectors as a function of rank, for VGG networks with frozen spatial filters trained on different classification tasks. The colormap range is defined in terms of the expected level of correlation between two random vectors (which is $1/\sqrt{d}$), so that this base level corresponds to the color white and statistically significant correlations (5 times this base level) correspond to the color black. The axis arrows indicate the effective rank of the corresponding weight covariance spectra. They show how the dimensionality of the learned subspaces increases with depth. **Top:** Networks are trained on ImageNet at $32 \times 32$ resolution with different labels. The first row indicates the similarity between learned encodings when only the random initialization is changed (indicated by the suffix "bis"). The subsequent rows show that the channel encodings emerging during training with true versus random labels are fundamentally different. However, the channel eigenvectors learned on different realizations of random labels are similar to each other, suggesting a consistent encoding strategy for random label tasks. **Bottom:** Networks are trained on various subsets of the CIFAR10, CIFAR100 and ImageNet datasets. The panels show that similar weight eigenvectors are consistently learned across datasets for a range of layers.

indicates the existence of some level of universality and two rather distinct encoding strategies which are shared across disjoint sets of tasks, when true or random labels are considered.

## 4 DISCUSSION

Does a universal encoding of natural images emerge in networks trained on different datasets? What does it look like? Answering these questions at the most fundamental level requires an investigation

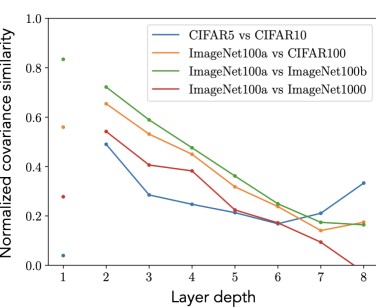

Figure 4: Normalized covariance similarities (eq. 6) for pairs of training tasks as a function of depth. The left panel confirms that the network learns different encodings when trained on true and random labels, but that this encoding does not depend on the realization of the random labels. The right panel quantifies the level of universality observed at each layer.

of the network weights. To do so, we have developed a procedure to compare the weights of networks rather than their representations. It extends the work of Guth et al. (2023) by including a space-channel weight factorization of standard CNN architectures, as well as dimensionality and similarity metrics which rely on optimal shrinkage of covariance eigenvalues. Our approach can be used to measure similarities between learned network weights and, equivalently, similarities between different datasets through their neural encodings. It can be used to define universality classes of datasets.

We have found that space and channel dimensions can be considered separately. In both cases, the relevant learned features are encoded through weight eigenvectors. Along spatial dimensions, we have evidenced a compact set of spatial filter eigenvectors which appears to be universal. It does not depend on filter size, layer depth, or training dataset over a wide range of settings. The same spatial filters also emerge when a network is trained on random labels. The learned channel eigenvectors also show signatures of universality. We show that they display a high level of correlation between datasets and tasks over an appreciable range of layers. Interestingly, we also show that when trained on random labels, VGG-type networks use a different encoding strategy. This second encoding however does not depend on the realization of the random labels, hinting that universality classes only depend on the amount of label noise. We do not attempt to characterize channel eigenvectors individually. In addition, assessing whether they are similar across layers is more challenging than along space and beyond the scope of this work. We do not yet know if similar functions are implemented at different layers. If so, one could develop architectures with weight-sharing across layers, greatly simplifying the learning process.

Our similarity metric can be used to identify trained networks with different or unexpected properties. It is now possible to measure the effect of architecture and training recipes on what is learned. For instance, it would be interesting for future work to look at the effects of adversarial training on the learned weight covariances as well as the type of task (classification versus generation). The metrics we introduce can also be computed at various points during training, and can thus be used to compare learning *trajectories* instead of just endpoints. Our results explain at a more fundamental level the success of transfer learning, self-supervised learning and foundation models. Instead of aiming at maximizing performance when training, one can also attempt to maximize the universality of the learned encoding and get closer to a foundation model. What are the architectural components that lead to increased universality of the learned features? Furthermore, an ensemble of trained networks can be viewed as a set of partial projections of such a foundation model. What would be the best strategy to define this universal encoding?

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

## A  VISUALIZATION OF LEARNED SPATIAL FILTERS

We show a portion of the spatial filters learned by VGG on ImageNet in Figure 5.

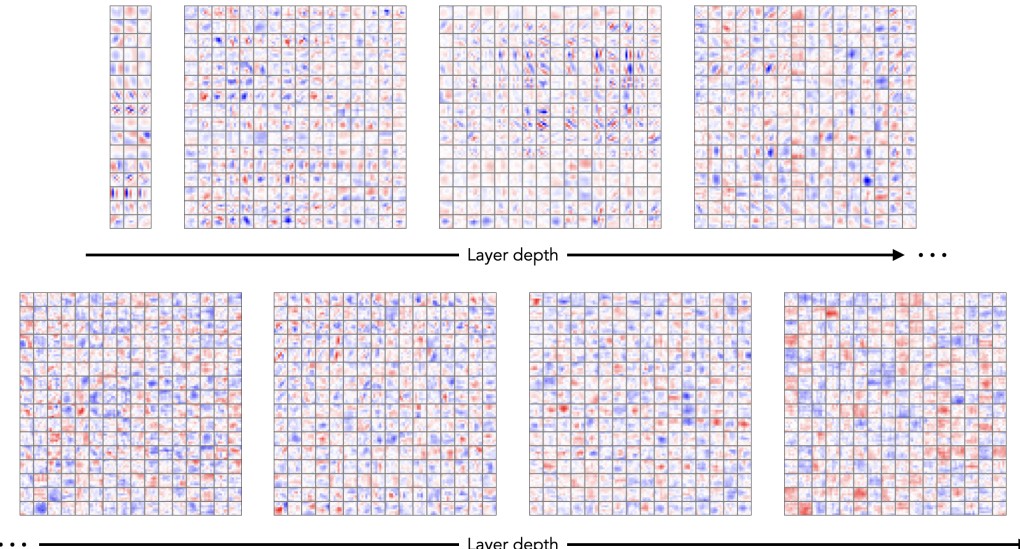

Figure 5: Visualization of the spatial filters learned by a VGG-11 network with 8 convolutional layers and filters of size $7 \times 7$. At each layer, we only show the filters corresponding to the first 16 input and output channels.

## B  EXPERIMENTAL DETAILS

### B.1  SIMPLIFIED VGG ARCHITECTURE

We perform several modifications on the VGG-11 architecture (Simonyan & Zisserman, 2015) with batch normalization layers (Ioffe & Szegedy, 2015) to simplify the analysis:

- We remove all learned biases from convolutional and linear layers.
- The batch normalization layers are positioned after the non-linearity (rather than before), and we remove their parameters learned by gradient descent (achieved with `affine=False` in the PyTorch library (Paszke et al., 2019)).
- We replace the MLP classifier with a single fully-connected linear layer (except in Figure 1 (a,b).

The first two modifications have a negligible effect on the classification performance of the trained networks. The third modification ensures that the vast majority of the parameters are in the convolutional layers so that our comparisons are as exhaustive as possible. However, our analysis could also be applied to the linear layers in the MLP classifier as-is.

### B.2  FACTORIZED ARCHITECTURES

In Section 3, we introduce factorized architectures. At each stage, the standard block JointConv–ReLU–BatchNorm (recall that we swapped the order between the non-linearity and the batch normalization layer) is replaced with DepthwiseConv–ReLU–BatchNorm–PointwiseConv. The depthwise convolution has $K = 10$ frozen (non-learned) filters. It can be implemented as a group convolution where the number of groups is equal to the number of input channels $C_{\text{in}}$ (resulting in $K\,C_{\text{in}}$ output channels). We use the first 5 universal eigenvectors $f_1, \ldots, f_5$ together with their opposites $-f_1, \ldots, -f_5$. Note that this equivalent to using an absolute value non-linearity concatenated with an identity skip-connection. The pointwise (or $1 \times 1$) convolution reduces the number of channels from

$K\,C_{\text{in}}$ to $C_{\text{out}}$ and is applied over channels only. It is thus a $C_{\text{out}} \times KC_{\text{in}}$ matrix, and corresponds to setting the kernel size of the convolution layer to $1$.

### B.3 TRAINING HYPERPARAMETERS

Network weights are initialized with i.i.d. samples from a uniform distribution (Glorot & Bengio, 2010) with so-called Kaiming variance scaling (He et al., 2015), which is the default in the PyTorch library (Paszke et al., 2019). Networks are trained for 90 epochs, with an initial learning rate of $0.005$ that is divided by 10 every 30 epochs. We use the SGD optimizer with a momentum of $0.9$ and no weight decay. We use classical data augmentations: horizontal flips and random crops for CIFAR10 and CIFAR100, and random resized crops of size $224$ and horizontal flips for ImageNet and subsets.

For the random label experiments in Figure 1 (b), following Zhang et al. (2021), we disabled data augmentation, removed batch normalization layers, and doubled training time (with learning decays being performed every 60 epochs). We also increased the initial learning rate to $0.02$. For the random label experiments in Figures 3 and 4, we kept data augmentation and batch normalization layers, and quadrupled the training time with an initial learning rate of $0.005$.

