# OpenReview forum: "On the universality of neural encodings in CNNs"
_ICLR.cc/2024/Workshop/Re-Align — ICLR 2024 Workshop Re-Align Poster_

### Official Review · Reviewer_cAhF · 2024-02-20
**A pA paper that presents some interesting tools for comparing network weights in CNNs along with experiments which support their value**

**Rating:** 2
**Fit:** 3
**Confidence:** 2

**Workshop Review:**

Strengths:
- This paper is mostly well-written. The diagrams communicate the experimental results well and are visually appealing.
- The reviewer agrees with the paper that the analysis of network weights has been understudied relative to the representations of data. This is a challenging problem and this work contributes what appear to be two useful analytical tools to this effort.
- The experiments are well chosen and interesting. In a maintrack version of the paper more experiments will be needed (see Weaknesses section below), but this is a good start.

Weaknesses:
- The one area for improvement is the mathematical descriptions. Trying to describe variants of convolutions in words is challenging and adding symbolic descriptions would be useful. For instance, this reviewer had to go back to Trockman, et. al. (2023) to confirm their understanding of the details in the spatial eigenvectors section. A symbolic description (with spatial and channel indices) would have made this unnecessary. The same observation holds for Section 3.1.
- The reviewer found the experiments on universal features to be compelling. To make the paper stronger, these should be further explored to better understand the extent to which they hold for different models, a broader swathe of datasets, and different training approaches. For instance, it would be very interesting to understand to what extent the universal features learned are impacted by details of the architecture of the network. Similarly, this reviewer was curious about the impact of dataset diversity vs dataset (image) resolution. ImageNet has higher resolution and is also arguably more diverse than CIFAR10, so it was hard to disentangle this from the present experiments.
- The weight analysis methods that are described in this paper all make sense. However, measuring and reasoning about complex and high-dimensional data like network weights is always challenging and the potential for pitfalls is ever present so it would probably make sense to include more sanity checking experiments in a future iteration of the paper. Specifically, 'are these experiments measuring what we think?'


Nitpicks:
- What are ‘IT spiking responses’?
- In Section 3.3, rank is displayed as a ‘value is indicated by an arrow overlaid on the rank axis.’ It took me a long time to figure out where this was. It might be good to have this better labeled in the figure.

Questions:
- The covariance values in Figure 3 vary from .9 to .99999. The reviewer wonders how meaningful differences in similarity on the order of .01 and smaller are. Is it likely that this is measuring something meaningful?

**Reason For Not Giving Higher Score:**

A more extensive exploration of the universal features experiments would make the work stronger. A clearer description of the methods in terms of the mathematics would make the work easier to understand.

**Reason For Not Giving Lower Score:**

This is an important research direction, and this reviewer feels that the proposed tools have the potential to shed light on representation learning in deep neural networks. The initial results on universal features are interesting and would likely be of interest to the community.

**Reviewer Domain:**

machine learning

---

### Official Review · Reviewer_XCQs · 2024-02-22
**A good paper, recommended for acceptance**

**Rating:** 2
**Fit:** 3
**Confidence:** 2

**Workshop Review:**

The paper motivates to give methods for comparing the weights between different models, rather than the conventional approach of comparing representations, where the authors employ the correlation coefficient between spatial covariance as the quantitative measure of similarity. They show that universal encoding pattern emerges in regards to both the spatial filters and channel weights over various datasets and tasks.

The paper is easy to follow and provides good presentations. I find the paper is quite interesting. Directly comparing weights between different models is challenging since no prior will be introduced during the comparison. And It is apparently beneficial to study the similarity metric between models. Additionally, this paper gives insights in regards to what could the model learn universally in different datasets.

Meanwhile, some concerns may also be raised. Firstly, as an empirically-focused study, the experiments could be further improved to better support the results. Secondly, I think the author may give also comparisons between models with same initialization. Also, the author may show the impact of techniques on this similarity. Thirdly, it is easy to come up with this interesting question: could we exchange the filters if they are quantitative similar? If so, it could be a solid evidence for the proposed metric.

**Reason For Not Giving Higher Score:**

N/A

**Reason For Not Giving Lower Score:**

This paper proposes a novel metric for comparing similarity between filters in CNNs. And this paper provides interesting insights regarding the universal encoding pattern between models and tasks. The authors present valuable findings, which can deepen our understanding towards the deep learning.

**Reviewer Domain:**

machine learning

---

### Official Review · Reviewer_vZSC · 2024-02-26
**Interesting paper, strong inclusion to workshop**

**Rating:** 2
**Fit:** 3
**Confidence:** 2

**Workshop Review:**

This paper introduces novel experiments with takeaways that will be relevant to the community.

## Strengths
- The authors introduce relevant questions (r.e. weight alignment between networks) and some relevant early results showing that the weights of vision models are often relatively aligned.
- The experiment described in section 3.2 and in Figure 5 is particularly interesting. The description and motivation for the alignment is likewise very clear.

## Weaknesses
- The experiments could use significantly better baselines/strengthening. For example, do the suggested results hold for models trained on data that lack relevant properties, e.g. non-images or for random initializations (and other relevant baselines)? The authors claim that the results explain transfer learning, what actually happens for models under transfer learning / finetuning? Do these results hold for models trained under different experimental settings, e.g. different optimizers, different datasets, architectures, and so on?
- Some of the claims are a bit strong.
   - In particular, in the abstract that the results "explain, at a more fundamental level, the success of transfer learning" does not seem clearly substantiated. The results in the paper give evidence, but the authors do not e.g. exhaustively show that transfer learning is not possible without this structure.
   - Likewise, the citations to claims in (Guth 2023) around universality and the relationship between task performance and effective rank are perhaps strong. For example, while the experiments might explain the models considered, they might not explain other crucial generalization properties of deep learning models [1].
   - I am unsure about the relevance to grokking. Under the definitions of grokking I have seen, the network abruptly finds a *generalizing* solution.
- Missing related work, e.g. [3] and references therein.

### Question
- How would things change in the results given in Fig. 5 if instead of rotations you used permutations, e.g. the permutation weight alignments used in [2]?
- The authors compare the weights of networks. How do such comparisons (e.g. in Fig 6) compare to related representation comparisons?

[1] Vyas, Nikhil, Yamini Bansal, and Preetum Nakkiran. "Empirical limitations of the NTK for understanding scaling laws in deep learning." Transactions on Machine Learning Research (2023).

[2] Ainsworth, Samuel K., Jonathan Hayase, and Siddhartha Srinivasa. "Git re-basin: Merging models modulo permutation symmetries." arXiv preprint arXiv:2209.04836 (2022).

[3] Huh, Minyoung, et al. "The low-rank simplicity bias in deep networks." arXiv preprint arXiv:2103.10427 (2021).

**Reason For Not Giving Higher Score:**

See weaknesses above.

**Reason For Not Giving Lower Score:**

See strengths above.

**Reviewer Domain:**

machine learning

---

### Decision · Program_Chairs · 2024-03-02

Accept (Poster)